# Improving Land Cover Classification Using Genetic Programming for Feature Construction

**João E. Batista** [1,*] , **Ana I. R. Cabral** [2] , **Maria J. P. Vasconcelos** [2] , **Leonardo Vanneschi** [3] and **Sara Silva** [1]

1 LASIGE, Faculty of Sciences, University of Lisbon, Campo Grande, 1749-016 Lisbon, Portugal; sgsilva@fc.ul.pt
2 Forest Research Centre, School of Agriculture, University of Lisbon, Tapada da Ajuda, 1349-017 Lisbon, Portugal; anaicabral@isa.ulisboa.pt (A.I.R.C.); mperestrelo@isa.ulisboa.pt (M.J.P.V.)
3 NOVA Information Management School (NOVA IMS), Universidade Nova de Lisboa, Campus de Campolide, 1070-312 Lisbon, Portugal; lvanneschi@novaims.unl.pt
* Correspondence: jebatista@fc.ul.pt

**Abstract:** Genetic programming (GP) is a powerful machine learning (ML) algorithm that can produce readable white-box models. Although successfully used for solving an array of problems in different scientific areas, GP is still not well known in the field of remote sensing. The M3GP algorithm, a variant of the standard GP algorithm, performs feature construction by evolving hyperfeatures from the original ones. In this work, we use the M3GP algorithm on several sets of satellite images over different countries to create hyperfeatures from satellite bands to improve the classification of land cover types. We add the evolved hyperfeatures to the reference datasets and observe a significant improvement of the performance of three state-of-the-art ML algorithms (decision trees, random forests, and XGBoost) on multiclass classifications and no significant effect on the binary classifications. We show that adding the M3GP hyperfeatures to the reference datasets brings better results than adding the well-known spectral indices NDVI, NDWI, and NBR. We also compare the performance of the M3GP hyperfeatures in the binary classification problems with those created by other feature construction methods such as FFX and EFS.

**Keywords:** genetic programming; evolutionary computation; machine learning; classification; multiclass classification; feature construction; hyperfeatures; spectral indices

## 1. Introduction

Since the establishment of the Warsaw Framework in 2013, remote sensing (RS) is recommended as an appropriate technology for monitoring and measuring, reporting, and verification (MRV) for countries reporting forest land cover and land cover change to the UNFCCC (https://unfccc.int/topics/land-use/resources/warsaw-framework-for-redd-plus (accessed on 15 November 2020)). However, many difficulties, from the availability of adequate in situ reference data to the spatial and temporal resolution of freely available satellite imagery and data processing power, have been hindering the operational use of this technology for MRV. Now, with the evolution of Earth observation systems (with provision of higher spatial and temporal resolution images) and with novel open-data distribution policies, there is an opportunity of applying machine learning (ML) to induce models that automatically identify land cover types in satellite images and improve the capacity for producing frequent and accurate land cover maps.

Previous ML work in classification of satellite imagery for land cover mapping has been successful. One simple practice that helps obtain good results is the inclusion of spectral indices as additional independent variables (also called attributes, or features, by the ML community) in the reference dataset. Spectral indices are combinations of reflectance values from different wavelengths that represent the relative abundance of certain terrain elements. They have been used by the RS community for a long time to enhance the identification of vegetation (e.g., NDVI [1]), water (e.g., NDWI [2]), burnt areas

(e.g., NBR [3]) and many other elements. Over the years, many indices were created and adapted to accommodate the particularities of different images. In the case of vegetation indices, this number is so vast that over one hundred of them were reviewed in [4].

Like indices, hyperfeatures are mathematical expressions that combine the original features of the data (the independent variables) with the goal of representing data properties that facilitate the learning of ML models. Spectral indices are, in fact, particular cases of hyperfeatures. Ideally, the hyperfeatures should be simple and meaningful, allowing the RS experts to easily understand the ML models that are based on them, or to directly use them in image analysis software to visualize what they represent.

Notwithstanding the success of ML methods when performing classification of satellite imagery, the reported results are often obtained by applying a model in the same images where it was trained (e.g., [5–7]), or in an image time series from the same location (e.g., [8–10]). Training models to be ready for use outside their training images is not a trivial task due to the radiometric variations between different images. These variations can arise from multiple sources, such as the difference in the angle of the solar incidence on the ground, the weather, the conditions of the terrain, the type of terrain, or the growth stage of the vegetation. Spectral indices are also sensitive to these variations, despite the efforts to increase their robustness.

Our goal is to improve satellite imagery classification by creating hyperfeatures that increase the performance of ML algorithms. In previous work [11], we used a genetic programming (GP) [12] classifier called M3GP [13] to evolve hyperfeatures that, when used instead of the original ones, were able to improve the accuracy of different ML algorithms in binary classification of images different from the ones used in training (although, for unseen data of the same images, there was no significant effect). GP is a powerful ML evolutionary algorithm that can produce readable white-box models. Successfully used for solving an array of problems in different scientific areas, GP is, however, still not well known in RS. The M3GP algorithm is a variant of standard GP that was originally developed as a multiclass classifier, but later used as a feature construction method for other algorithms, both for classification and for regression [11,14,15]. Creating hyperfeatures from one image and using them for classifying a different image falls under the area of transfer learning [15], which attempts to use knowledge from one problem to solve another similar problem.

In this work, we perform a thorough study of the effects of adding M3GP-evolved hyperfeatures to the reference datasets. We test our approach on several datasets from different images in two types of problems that have been tackled several times over the last decades, the binary classification of burnt areas [16–22], and the multiclass classification of land cover types [23–27]. The images used in our study cover several different regions over developing countries: Angola, Brazil, Democratic Republic of the Congo, Guinea-Bissau, and Mozambique. We add the evolved hyperfeatures to the reference datasets and analyze the differences in the generalization ability of different ML algorithms when tested on unseen data from the same images. Three common state-of-the-art algorithms are tested, namely decision trees [28], random forests [29] and XGBoost [30]. We also perform the same experiments when adding spectral indices instead of the hyperfeatures, comparing the results. The selected indices are the popular NDVI, NDWI, and NBR. For the binary classification problems, we also compare our results with the ones obtained when adding hyperfeatures created by two different feature construction methods, EFS [31] and FFX [32].

## 2. Related Work

Feature engineering is an essential step in the knowledge discovery process and one of the keys to success in applied ML. The features used to induce a data model can directly influence the quality of the model itself and the results that it can achieve. Feature engineering can be broadly partitioned into feature selection and feature construction. According to Liu and Motoda [33], feature selection is a process that chooses a subset of features from the original data variables, so that the feature space is optimally reduced according to a certain criterion, whereas feature construction/extraction (also called feature generation, feature

learning, or constructive induction) is a process that creates a new set of hyperfeatures from the original data. Feature construction typically combines existing variables into more informative hyperfeatures. Both feature selection and feature construction attempt to improve model performance and can be used in isolation or in combination.

Feature construction, the focus of this work, has been widely studied in the last two decades. Recent surveys can be found in [34–36], while the book [37] gives an in-depth presentation of the area. In all these references, the importance of EC as an effective method for feature construction is asserted, together with other feature construction methods such as the ones based on decision trees, inductive logic programming, and clustering. A recent survey of EC techniques for feature construction can be found in [38]. Among the different EC flavors, GP is probably the one that has been used more often and more successfully. Indeed, GP is particularly suited for feature construction because it naturally evolves functions of the original variables. The versatility offered by the user-defined fitness function of GP allows the user to choose among several possible criteria for evolving new hyperfeatures. Additionally, the fact that the evolved hyperfeatures are, in principle, readable and understandable can play an important role in model interpretability. Several existing GP-based methods for feature construction are discussed in [39,40], and a deep analysis of previous work can be found in [15], where GP-constructed features are used for transfer learning.

Among the large set of feature construction methods available, in this paper, we use M3GP [13] as our method of choice and two others for comparison purposes: the non-EC method FFX [32] and the EC method EFS [31]. In the remainder of this section, will focus on GP-based feature construction, including applications, and on feature construction and GP in the context of RS.

### 2.1. Feature Construction with Genetic Programming

Among the several previous contributions in which GP was used for feature construction, Krawiec has shown that classifiers induced using the representation enriched by GP-constructed hyperfeatures provide better accuracy on a set of benchmark classification problems [41]. Krawiec et al. have also used GP in a coevolutionary system for feature construction [42,43].

The use of GP for feature construction was later deeply investigated by Zhang et al. [44–48]. For instance, in [44], a GP approach was proposed that, instead of wrapping a particular classifier for single feature construction as in most of the existing methods, it used GP to construct multiple features from the original variables. The proposed method used a fitness function based on class dispersion and entropy, and thus was independent of any particular classification algorithm. The approach was tested using decision trees on the new obtained dataset and experimentally compared with the standard decision tree method, using the original features. The results showed that the proposed approach outperforms standard decision trees on the studied test problems in terms of the classification performance, dimension reduction, and the learned decision tree size. Several years later, in [45], GP was used for both feature construction and implicit feature selection. The work presented a comprehensive study, investigating the use of GP for feature construction and feature selection on high-dimensional classification problems. Different combinations of the constructed and/or selected features were tested and compared on seven high-dimensional gene expression problems, and different classification algorithms were used to evaluate their performance. The results indicated that the constructed and/or selected feature sets can significantly reduce the dimensionality and maintain or even increase the classification accuracy in most cases. In [46], previous GP-based approaches for feature construction were extended to deal with incomplete data. The results indicated that the proposed approach can, at the same time, improve the accuracy and reduce the complexity of the learnt classifiers. While until a few years ago, GP-based feature construction had been applied mainly to classification, in [47], it was applied with success to symbolic regression, thus giving a demonstration of the generality of the approach. In [48], the authors investigate

the construction of sets of hyperfeatures using GP-based approaches. One of the most interesting results showed that multiple-feature construction achieves significantly better performance than single-feature construction. Consistently with that result, the method presented in this paper uses GP to construct multiple features.

It should be noted that the use of GP for feature construction has been explored for some time, as surveyed in [40]. Although the most common approach to multiclass classification problems used to be splitting a classification problem with $n$ classes into $n$ binary classification problems, and evolving one hyperfeature for each class [49,50], some methods create several hyperfeatures to separate the classes within the feature space. In this category, the survey includes works that converted the datasets into hyperdatasets using exclusively the evolved hyperfeatures [41], and works that include the original hyperfeatures in the hyperdataset [51] (similarly to our work).

GP-based feature construction methods have been used with success in several real-life applications. For instance, the authors of [52] proposed a novel method for breast cancer diagnosis using the features generated by GP. A few years later, in [53], GP-based feature construction was used for improving the accuracy of several classification algorithms for biomarker identification. In [54], a method to find smaller solutions of equally high quality compared to other state-of-the-art GP approaches was coupled with a GP-based feature construction method and applied to cancer radiotherapy dose reconstruction. One year later, in [55], GP-based feature construction was successfully applied to the classification of ten different categories of skin cancer from lesion images. Interestingly, while the application tackled in [54] is a symbolic regression problem, the one in [55] is a multiclass classification problem, thus confirming that the GP-based feature construction approach can be successfully applied to both types of problems. Finally, in [56], GP-based feature construction was extended for the first time to experimental physics. In particular, to be applicable to physics, dimensional consistency was enforced using grammars. The presented results showed that the constructed hyperfeatures can both significantly improve classification accuracy and be easily interpretable.

### 2.2. Feature Construction and Genetic Programming in Remote Sensing

In the RS domain, many techniques have been used to extract features from satellite images. These features include statistical descriptors, obtained by the gray level co-occurrence matrix (GLCM) and other methods [57]; features of interest, such as known structures (e.g., buildings, roads), using deep learning [58]; sets of generic features, using principal component analysis (PCA) [59]; and even temporal features, using the continuous change detection and classification (CCDC) algorithm [60].

GP-based algorithms, mainly the standard GP algorithm, have been previously used in the area of RS in tasks such as the creation of vegetation indices [61], the detection of riparian zones [62] and the estimation of soil moisture [62,63], the estimation of canopy nitrogen content at the beginning of the tasseling stage [64], the estimation of chlorophyll levels to monitor the water quality in reservoirs [65], the prediction of soil salinity by estimating the electrical conductivity on the ground [66], and also in geoscience projects reviewed in [67].

The expressions obtained by the GP-based algorithms can be used in transfer learning by exporting them to datasets under the form of hyperfeatures, in the attempt to improve the performance of ML algorithms. Our work continues to develop this kind of application, which was already explored in the area of RS using EC-based algorithms [68,69] and specifically GP-based algorithms [11,62].

## 3. Materials and Methods

### 3.1. Datasets and Study Areas

The datasets used in this work are meant to train ML models to classify burnt areas and land cover types on a pixel-level. We use a total of nine datasets, obtained from Landsat-7, Landsat-8, and Sentinel-2A satellite images. The characteristics of these images

and datasets are summarized in Tables 1 and 2, and their associated geographic locations are highlighted in Figure 1.

**Table 1.** Summary of the datasets used.

| Dataset | Ref. | Country | Scene Identifier Path/Row | Acq. Date DD/MM/YYYY | No. Images | Satellite | KGCS |
|---|---|---|---|---|---|---|---|
| Ao2 | [a] | Angola | 177/67 | 09/07/2013 | 1 | LS-8 | Cwa |
| Br2 | [70] | Brazil | 225/64 | 28/02/2015 | 1 | LS-8 | Af, Am |
| Cd2 | [70] | DR Congo | 175/62 | 08/06/2013 | 1 | LS-8 | Aw |
| Gw2 | [70] | Guinea-Bissau | 204/52 | 13/05/2002 | 1 | LS-7 | Am, Aw |
| IM-3 IM-10 | [b] [71] | Guinea-Bissau | 203/51, 52 204/51, 52 205/51 | From: 02/01/2010 To: 01/04/2010 | 17 | LS-7 | Am, Aw |
| Ao8 | [72] | Angola | 182/64, 65 | 18/06/2016 | 2 | LS-8 | Aw |
| Gw10 | [c] | Guinea-Bissau | 204/51, 52 205/51 | 01/03/2019 24/03/2019 | 3 | LS-8 | Am, Aw |
| Mz6 | [73] | Mozambique | Entire Country ( 122 S-2A tiles ) | From: 19/02/2016 To: 06/10/2016 | 2806 [d] | S-2A | Am, Aw, BSh, Cwa, Cwb, Cfa |

[a] There is no reference paper for this dataset. [b] This is a subdataset, obtained by extracting three forest classes from the IM-10 dataset. [c] The reference paper for this dataset is under review. [d] An approximation obtained by considering that the S-2A mapped every tile of Mozambique once every 10 days for 230 days.

**Table 2.** Summary of the datasets used.

| Dataset | Classes (No. Pixels) | | | No. Classes | No. Bands No. Features | Total Pixels |
|---|---|---|---|---|---|---|
| Ao2 | Burnt (1573) | Non-Burnt (2309) | | 2 | 7 | 3882 |
| Br2 | Burnt (2033) | Non-Burnt (2839) | | 2 | 7 | 4872 |
| Cd2 | Burnt (877) | Non-Burnt (1972) | | 2 | 7 | 2849 |
| Gw2 | Burnt (1101) | Non-Burnt (3430) | | 2 | 7 | 4531 |
| IM-3 | Savanna Woodland (114) | Dense Forest (68) | Open Forest (140) | 3 | 6 | 322 |
| IM-10 | Agriculture/Bare Soil (950) Grassland (75) Savanna Woodland (1626) Water (908) | Burnt (77) Mangrove (1240) Sand (166) | Dense Forest (524) Open Forest (723) Mud (509) | 10 | 6 | 6798 |
| Ao8 | Agriculture/Bare Soil (73) Forest (662) Savanna Woodland (598) | Burnt (301) Grassland (12) Water (152) | Clouds (332) Urban (53) | 8 | 10 | 2183 |
| Gw10 | Agriculture/Bare Soil (449) Grassland (16) Savanna Woodland (1308) Wetland (389) | Burnt (157) Mangrove (1383) Sand (50) | Dense Forest (62) Open Forest (646) Water (620) | 10 | 7 | 5080 |
| Mz6 | Agriculture/Bare Soil (33611) Urban (4194) | Forest (63190) Wetland (35673) | Grassland (28406) Other (25128) | 6 | 10 | 190202 |

### 3.1.1. Datasets

From the Landsat-7 images, we have one binary classification dataset (Gw2) and two multiclass classification datasets (IM-10 and IM-3). The IM-3 dataset was built, in previous work, from IM-10 by extracting only the pixels classified in situ from the three forest land cover types that ML models failed to correctly discriminate. These images were both obtained over Guinea-Bissau.

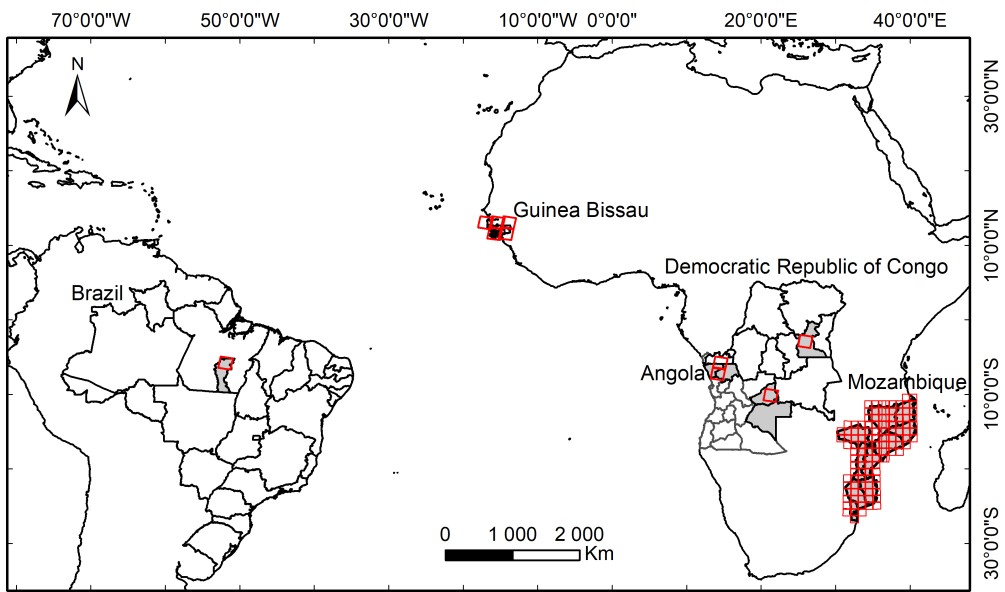

**Figure 1.** Location of the study areas, in red, in Brazil (Br2), Guinea-Bissau (Gw2, Gw10, IM-3, IM-10), Democratic Republic of Congo (Cd2), Angola (Ao2, Ao8), and Mozambique (Mz6) in South America and Africa continent.

From the Landsat-8 images, we have three binary classification datasets and two multiclass classification datasets. The binary classification datasets have the objective of training models to identify burnt areas, by classifying each pixel as "burnt" or "nonburnt". These three datasets were obtained from satellite images over Angola (Ao2), Brazil (Br2) and Democratic Republic of the Congo (Cd2). The multiclass classification datasets have the objective of training models to correctly classify each pixel as one of several different land cover types. These two datasets were extracted from satellite images over Angola (Ao8) and Guinea-Bissau (Gw10).

Lastly, from the Sentinel-2A satellite images, we have one multiclass classification dataset that was extracted from several satellite images from the entire country of Mozambique (Mz6). These images were obtained through 2016, between 19 February and 6 October [73].

### 3.1.2. Study Areas

In terms of size of the classified areas, the pixels used in the Landsat satellite images consist of 900 m$^2$ areas and those used in the Sentinel-2A satellite images consist of 400 m$^2$ areas. As such, the classified areas can be calculated from Table 2. Next, we describe the climate (according to the Köppen–Geiger classification system (KGCS) [74]) and vegetation in each of the study areas:

**Brazil:** The study area of the Br2 dataset is located in eastern Amazonia, in southeastern Pará, Brazil. According to the KGCS, the climate in this image is classified as equatorial monsoon (Am) and equatorial rainforest, fully humid (Af) in the north and south sections, respectively. This area is drier than central and western Amazonia, with annual rainfall between 1500 mm and 2000 mm and average temperatures ranging from 23 °C to 30 °C. The vegetation in this image ranges from lowland Amazon forest in the north through submontane dense and open forests in the south [70].

**Guinea-Bissau:** The study area of the Gw2, IM-3, IM-10, and Gw10 datasets is located in Guinea-Bissau, West Africa. According to the KGCS, the climate in this area is classified as Am and equatorial savanna with dry winter (Aw) within the coastal and interior areas, respectively. This area is characterized by having a marshy coastal plain with a dry to moist (North to South) tropical climate. There are two marked seasons, a dry season between November and May, and a wet season between June and October. Total annual rain values vary from 1200 to 1400 mm in the Northeast region, and from 2400 to 2600 mm in the

Southwest region. The monthly average temperature ranges from 25.9 °C and 27.1 °C. The vegetation consists of mangroves on the coast and gradually becomes composed of mainly dry forest and savanna inland [70].

**Northern Angola:** The study area of the Ao8 dataset is located in the Zaire province, northern Angola. According to the KGCS, the climate in this region is classified as Aw with a mean annual rainfall near 1300 mm, distributed in two periods separated by a short dry spell. The monthly average temperature ranges from 20.5 °C and 24.9 °C. The vegetation is mainly savanna scrublands and some dense humid forests mostly located along rivers, creeks, and gullies. There are anthropic forests composed by native species and mango, cola, safou, avocado, citrus, and guava trees in ancestral settlements, abandoned due to forced relocation along the main roads by the colonial administration [75].

**Eastern Angola:** The study area of the Ao2 dataset is located in Lunda Sul, Eastern Angola. According to the KGCS, the climate in this area is classified as Warm temperate climate with dry winter and hot summer (Cwa) and a mean annual rainfall near 1300 mm, distributed between October and April and a dry season from May to September. The monthly average temperature ranges from 20.0 °C and 24.4 °C. The vegetation is mainly dominated by woody and shrub savannas and gallery forests essentially located along the valleys of the great rivers [76].

**Democratic Republic of Congo:** The study area of the Cg2 dataset is located in the central-eastern Democratic Republic of Congo. According to the KGCS, the climate in this area is classified as Aw with a mean annual rainfall near 1600 mm. There are two distinct seasons, a dry season (with temperatures ranging between 18 °C and 27 °C) from June to August, and a rainy season (with temperatures ranging between 22 °C and 33 °C) from September to May. The vegetation is characterized by a congolian lowland forest in the north to miombo woodlands in the south. In the southwestern region, the population pressure had conducted to the degradation of the miombo woodlands [70].

**Mozambique:** The study area of the Mz6 dataset includes the entire country of Mozambique. According to the KGCS, the climate is classified as Aw in the coastal area and near the Zambezi river; as Cwa in the interior, at the north and the west of the Zambezi river; as warm temperate climate with dry winter and warm summer (Cwb) near Lichinga and west of Chibabava; as hot semiarid (BSh) in the interior in south Mozambique and east of Mungári and Derre, and as hot desert (Bwh) in the area between Dindiza and the frontier between Mozambique and Zimbabwe, south of the Save river. It has a wet season from October to March and a dry season from April to September. The lowest average rainfall (300–800 mm/year) occur in the interior southern regions and the highest average rainfall (over 1200 mm/year) occur in the area around Espungabera. The average temperatures are the highest along the coast in the northern regions (with temperatures ranging between 25 °C and 27 °C in summer and between 20 °C and 23 °C in winter) and in the southern regions (with temperatures ranging between 24 °C and 26 °C in summer and between 20 °C and 22 °C in winter), while the high inland regions have cooler temperatures (with temperatures ranging between 18 °C and 20 °C in summer and between 13 °C and 16 °C in winter). The northern areas are predominantly occupied by miombo woodlands and the western and southern borders by Zambezian and Mopane woodland. The most widespread vegetation in the north coast is the Zanzibar–Inhambane forest mosaic, followed by the African mangroves and the Maputaland forest mosaic in the southeast coast [73,77,78].

*3.2. Methodology*

The core of this work is to expand the reference datasets with hyperfeatures that improve the performance of different ML methods. Figure 2 illustrates the process of obtaining and using such hyperfeatures. As usual, the reference dataset is split in two datasets, one for training the classifiers, called the training set, and one for testing the classifiers on unseen data, called the test set. Based only on the training set, the feature construction algorithm creates a set of hyperfeatures that are used to expand the reference dataset in both training and test sets. The expanded training set is used by the classification

algorithm to obtain a trained classifier that is applied to both (expanded) training and test sets in order to report the performance in terms of learning and generalization, respectively.

A small deviation to this process has been made for the datasets IM-3 and IM-10, where the hyperfeatures used to expand the IM-10 datasets where obtained in the training data of its subset IM-3, and not the training data of the complete IM-10. The goal of this deviation was to check whether a larger dataset could also benefit from hyperfeatures obtained in a much limited context (results reported in Section 4.3).

As feature construction algorithms, we use M3GP and compare it with FFX and EFS, all described below. As classification algorithms, we use decision trees (DT), random forests (RF), and XGBoost (XGB), also briefly described below. The number and complexity of the created hyperfeatures is not predefined, but automatically determined by the feature construction algorithm.

We also experiment with expanding the reference datasets with the NDVI, NDWI, and NBR indices, instead of performing feature construction. These indices were selected from the RS literature as being helpful to the ML algorithms for separating vegetation, water, and burnt classes, since these elements are present among the pixels used in the datasets.

Each experiment is performed 30 times for each possible trio of reference dataset, feature construction algorithm, and classification algorithm (with the exception of the EFS and FFX algorithms, which are only used in binary classification datasets), each time with a different random split of the reference dataset in training and test sets. In other words, and limiting the explanation to Figure 2, our experimental process follows these steps:

**Splitting the Dataset:** The reference dataset is split randomly into training (70% of the pixels) and test sets (remaining 30%), stratified by class;

**Creating and Adding Hyperfeatures/Indices:** The training set is used by a Feature Construction algorithm to create a new set of hyperfeatures, and the training and test sets are then extended using these hyperfeatures, or the indices;

**Training and Testing a Classifier:** a classifier is trained using the extended training set and tested on the extended test set, providing the final results.

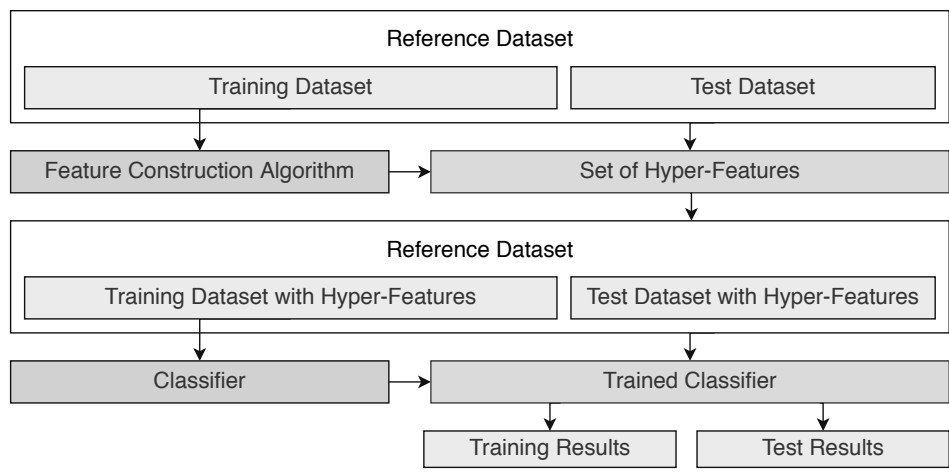

**Figure 2.** Representation of the methodology adopted to obtain and use hyperfeatures.

### 3.3. Feature Construction Algorithms

We use three different methods for feature construction. Our method of choice is the M3GP algorithm, because of the interpretability of the hyperfeatures it creates, and because it can evolve hyperfeatures for multiclass classification problems. For comparing our M3GP results with the results of other evolutionary and non-evolutionary methods, we selected the EFS and FFX algorithms, due to their running speed, availability of the authors' implementations, and number of citations. However, EFS and FFX are focused on regression problems, rather than classification problems. They are easily adapted to binary classification by defining a threshold separating the two classes, but there is no

easy adaptation for multiclass problems, the reason why we test them only on the binary classification datasets.

**M3GP algorithm:** Multidimensional multiclass GP with multidimensional populations (M3GP) is a GP-based algorithm that evolves a set of hyperfeatures that convert the original feature space into a new feature space, guided by a fitness function that measures the performance of a classifier in the new feature space. The M3GP is an all-in-one algorithm that both creates the hyperfeatures and uses them for solving both regression and classification problems. The inner workings of this algorithm are explained below, in Section 3.3.1.

**EFS algorithm:** Evolutionary feature synthesis (EFS) is an evolutionary algorithm that uses pathwise LASSO [79] regression to optimize multiple linear regression models that are extended for nonlinear relationships between features. This extension is made using functions such as *cos*, *sin*, and *log*, as well as functions with several inputs, e.g., multiplication of variables. This regression tool can produce a set of interpretable hyperfeatures in seconds.

**FFX algorithm:** Fast function extraction (FFX) is a deterministic algorithm that applies pathwise regularized learning [80] to a large set of generated nonlinear functions to search for a set of hyperfeatures with minimal error. Although the hyperfeatures observed are simple, this algorithm generates hundreds of hyperfeatures, which leads us to consider the final model noninterpretable.

### 3.3.1. The M3GP Algorithm

The M3GP [13] is a GP-based algorithm that evolves models similar in structure to the models of standard GP [12]. Standard GP represents each model as a parse tree, to be read depth-first, where the terminal nodes are features and the nonterminal nodes are operators that combine features (e.g., arithmetic operators such as multiplication, subtraction, etc.). The main difference between the models evolved by M3GP and the ones of standard GP is that in M3GP, each model is not a single tree, but a set of trees, as exemplified in Figure 3. These trees are what we call hyperfeatures and are evolved using the steps shown in Figure 4 and explained here:

**Initialization:** The M3GP algorithm initializes its population of models as single, randomly generated trees. Therefore, in the beginning of the evolutionary cycle, each individual is a simple model consisting of a single hyperfeature.

**Evaluation:** Each individual of the population is evaluated by the following procedure. The *n* hyperfeatures are used to convert the original features into a new *n*-dimensional dataset. The fitness of the model is then calculated by applying a fitness function to this new (hyper)feature space. This fitness function rewards the individuals whose set of hyperfeatures creates a space where the different classes are more easily separable. In the original M3GP algorithm, the fitness was the overall accuracy of the Mahalanobis distance classifier (described below), but in the current implementation, we use the weighted average of *F*-measures (WAF) instead of the overall accuracy, (although we still use the overall accuracy to assess performance), for its robustness to class imbalance, especially in multiclass classification.

**Stopping Criteria:** After the population is evaluated, the algorithm decides whether to stop the evolution or not. The most common stopping criteria are related to the number of generations/iterations already done, and to the quality of the best model achieved (in terms of accuracy or any other metric). In the current implementation, the evolution stops when 50 generations are completed or when one individual achieves 100% accuracy on the training set, whichever occurs first. If the evolution does not stop, a new generation of models is created, following the steps described next.

**Selection:** The parents of the next generation are selected using the tournament method. To select each parent, the algorithm randomly selects a small group of models and, out of these models, chooses the best. The tournament method is able to maintain enough selective pressure to choose mostly the best individuals, thus promoting the propagation of

their good traits in the next generation, while allowing also the bad ones to become parents, thus avoiding the loss of genetic diversity that would stagnate the evolution.

**Breeding:** After selecting models to act as parents, each new model is created either through a mutation of one parent or through a crossover of two parents. When using a mutation genetic operator, the parent can either: create a new, randomly generated tree and add it to its set of hyperfeatures; randomly select one of its hyperfeatures and remove it (if it contains more than one hyperfeature); or modify one of its hyperfeatures by replacing one of its branches with a new, randomly generated tree. When using a crossover genetic operator, the parent models can swap either branches or entire hyperfeatures between each other. Unlike the mutation genetic operator, the crossover results in two offspring.

After a new population has been created, the algorithm returns to the evaluation step.

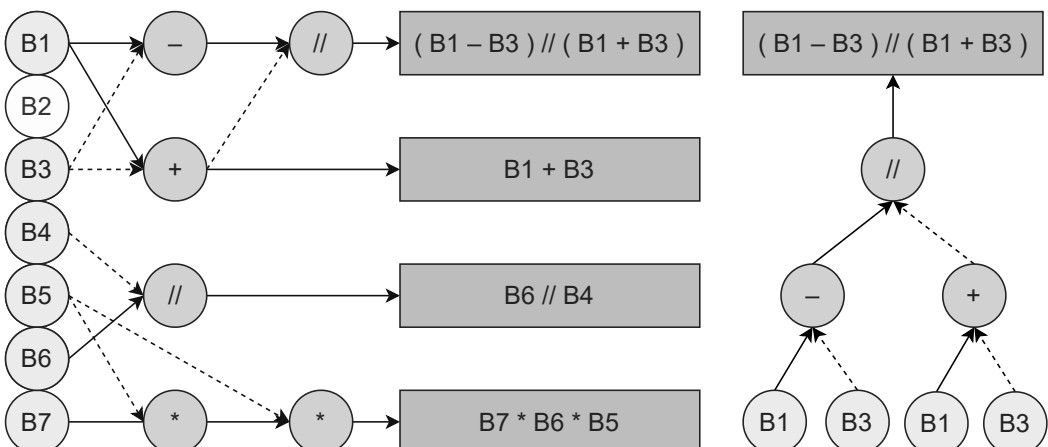

**Figure 3.** Example of an M3GP model that uses six of the seven available features to build four hyperfeatures (**left**) and a single hyperfeature (**right**). The solid and dashed lines indicate the first and second variables used by the operators. // is a division operator protected against division by zero.

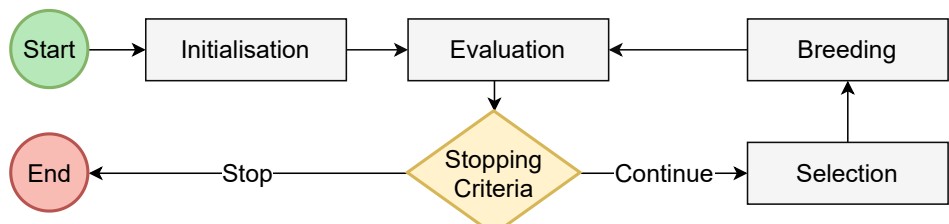

**Figure 4.** Evolutionary cycle used by M3GP.

### 3.4. Classification Algorithms

Four different classifiers are used in this work: MD, DT, RF, and XGB. The MD classifier is used only as part of M3GP, but the other three are used to independently test the effectiveness of the indices and hyperfeatures added to the reference datasets.

**Mahalanobis distance classifier:** The MD classifier is a nonparametric supervised cluster-based algorithm that classifies a data point by associating it with the closest cluster centroid using the Mahalanobis distance, where a cluster is defined as a set of pixels belonging to the same class.

**Decision tree classifier:** The DT algorithm is a nonparametric supervised algorithm that infers simple decision rules from the training data. This algorithm can be used in both classification and regression problems.

**Random forest classifier:** The RF algorithm is an ensemble algorithm that uses a set of DT to solve both classification and regression problems by assigning each data point to

the majority vote of all DT in classification problems or to the average of the prediction of all DT in regression problems.

**XGBoost classifier:** The XGB algorithm is a DT-based ensemble algorithm that uses an optimized gradient boosting to minimize errors. This algorithm can be used in both classification and regression problems.

### 3.5. Tools and Parameters

All the experiments involving M3GP are performed using our own implementation of the M3GP algorithm (Python implementation available at https://github.com/jespb/Python-M3GP), which includes the implementation of the MD classifier. The DT and RF implementations belong to the sklearn Python library [81] and the XGB implementation belongs to the xgboost Python library [30]. The EFS implementation (EFS project website: http://flexgp.github.io/efs/) is provided by the authors in their paper [31] and the FFX implementation (Python implementation available at https://github.com/natekupp/ffx) belongs to the ffx Python library.

The parameter settings used in this work are the standard within the ML community, with the main parameters and variations specified in Table 3. The EFS, FFX, and M3GP algorithms used the same parameters as those used by the authors in their respective papers. The variations in this work include our implementation of the M3GP using the WAF of the MD classifier (untied with the number of hyperfeatures and then with the total size of the model) as fitness, rather than the accuracy, and only pruning the final individual, for consistency with previous work that had this variation [11]. Every run using the DT, RF, and XGB classifiers used the default parameters of their respective implementations, except for the XGB runs in the Mz6 dataset. In this dataset, the XGB was unable to obtain perfect training accuracy with the default maximum depth for its models. As such, the maximum depth was increased from 6 to 20.

**Table 3.** Main parameters and variations used in the experiments.

| | |
|---|---|
| General: | |
| Runs | 30 |
| Training Set | 70% of the samples of each class |
| Statistical Significance | $p$-value $< 0.01$ (Kruskal–Wallis $H$-test) |
| M3GP: | |
| Stopping Criteria | 50 generations or 100% training accuracy |
| Fitness | WAF (Weighted Average of $F$-measures) |
| Pruning | Final individual |
| XGBoost: | |
| Maximum Depth | 20 in the Mz6 dataset and 6 (default) in the other datasets |

## 4. Results and Discussion

We start this section by presenting the results and hyperfeatures obtained by running M3GP by itself on all the datasets. Then, we discuss the interpretability of the hyperfeatures and the popularity of the different satellite bands in the solutions proposed by M3GP. Next, we compare the overall accuracy, and class accuracy, obtained when running the DT, RF, and XGB algorithms on the original datasets and on the datasets expanded with indices or hyperfeatures. Regarding the class accuracy, we present the results only for XGB because the results for DT and RF were similar in terms of the relationship between the classes, and therefore would not bring any new information.

The presentation of the results is split into three categories: binary classification datasets (Ao2, Br2, Cg2, and Gw2), regarding the detection of burnt areas; discrimination of similar classes (IM-3 and IM-10), regarding the separation of forest types; and discrimination of all classes (Ao8, Gw10, and Mz6), regarding the separation of different land cover types.

We present the results in tables, boxplots, and confusion matrices. On the tables, each overall accuracy value is the median obtained in the 30 runs. The confusion matrices, rather than showing the class accuracy, show the difference (in percentage of pixels) between using hyperfeatures (or indices) and using the original dataset, to facilitate the identification of the effect produced by the hyperfeatures. Statistical significance is determined with the nonparametric Kruskal–Wallis $H$-test (from the scipy Python library) at $p < 0.01$.

### 4.1. M3GP Performance and Hyperfeature Analysis

Although we use M3GP as a feature construction method for other ML algorithms, M3GP can perform binary and multiclass classification by itself, as described in Section 3.4. While using M3GP to evolve the hyperfeatures, we have registered the accuracy values it achieved in each dataset, presented in Table 4. Although the accuracy is high, it is generally worse than the accuracy achieved by the other ML algorithms we used, and therefore we will not refer to the results of standalone M3GP again.

In terms of interpretability of the evolved hyperfeatures, in Table 4, we report the number of hyperfeatures and their median size (with minimum and maximum values, between parentheses). While the number of evolved hyperfeatures seems to depend heavily on the number of classes of the problem, the median average size of each hyperfeature tends to be higher for the binary datasets, where a large dispersion of values is observed.

To exemplify the variety of different sets of evolved hyperfeatures, we picked three examples. On the first two examples, a single hyperfeature was evolved, but with different sizes. Both were evolved for the Gw2 dataset and obtained perfect test accuracy on the respective runs. The third example is a set of 16 hyperfeatures that were evolved in a run for the Ao8 dataset and obtained median test accuracy. This variety of hyperfeatures can be seen in Equations (1)–(7). Note that $Bn$ refers to the $n$th band of the satellite. As we can see, the M3GP algorithm can generate hyperfeatures that are as simple as (and perfectly equal to) the original features themselves (Equations (3)), hyperfeatures that are simple enough to be interpreted (Equations (1), (4) and (5)), and hyperfeatures that need to be decomposed for a proper analysis of the expression (Equations (2), (6) and (7)).

Looking at Table 4 and the examples of hyperfeatures in Equations (1)–(7), we can state that, although the M3GP sometimes produces complex hyperfeatures, the general case seems to be the production of interpretable hyperfeatures. While this work focuses exclusively on datasets from the RS domain, the same tendency regarding interpretability was already observed in the original M3GP paper [13], which used datasets from a much wider range of domains.

**Table 4.** The median training and test overall accuracy, size, number of hyperfeatures, and average size of the hyperfeatures obtained by the M3GP models in 30 runs in each dataset.

|  | Ao2 | Br2 | Cg2 | Gw2 | IM-3 | IM-10 | Ao8 | Gw10 | Mz6 |
|---|---|---|---|---|---|---|---|---|---|
| **Accuracy** | | | | | | | | | |
| **Training** | 1.000 | 0.992 | 0.993 | 1.000 | 0.996 | 0.932 | 1.000 | 0.988 | 0.620 |
| **Test** | 0.999 | 0.990 | 0.993 | 1.000 | 0.948 | 0.916 | 0.983 | 0.971 | 0.620 |
| Hyperfeatures | | | | | | | | | |
| Number | 5 (3–8) | 8 (1–15) | 4 (3–8) | 2 (1–3) | 8.5 (5–13) | 23 (17–29) | 18 (14–21) | 21 (15–23) | 14.5 (12–17) |
| Avg. Size | 14 (9–28) | 14 (6–39) | 23 (6–40) | 11 (2–43) | 11 (5–22) | 10 (8–13) | 10 (5–14) | 8 (6–12) | 11 (5–17) |

**Gw2, Run#11, 1 Hyperfeature:**

$$\frac{B5\,(B3 + B5)}{B7 + B4 + 1} \tag{1}$$

**Gw2, Run#26, 1 Hyperfeature:**

$$\frac{B2^2\,B3\,B4\,B5\,B6 \;-\; B2^2\,B4^2\,B5 \;+\; B2\,B3^2\,B5^2\,B6 \;-\; B2\,B3\,B4\,B5^2 \;-\; B3\,B4^3\,B7}{B4^2\,B7\,(B2\,B4 \;+\; B3\,B5)} \tag{2}$$

**Ao8, Run#18, 16 Hyperfeatures:**

$$B3 \qquad B5 \qquad B6 \qquad B10 \qquad B11 \qquad (3)$$

$$B9 - B2 \qquad \frac{B3}{B5} \qquad \frac{B1}{B2\,B7^2} \qquad B5 - B6 + B9 - 2\,B10 \qquad \frac{B2\,B9}{B2\,B11 - B10 - B2\,B5} \qquad (4)$$

$$\frac{(B1 + B4 - B10)\,(B3 + B9)}{B6} \qquad B6\,B9 - B1\,B2 - B1 + B3 + B6 - B9 \qquad (5)$$

$$\frac{B9\,(B9 - B11)}{B7\,(B3\,B6 + B9 - B11)} \qquad (B4 + B10 - \frac{B1^2}{B5 - B9})\,(2\,B2 + \frac{B3}{B4} - B4 + B5 + B10) \qquad (6)$$

$$\frac{B7\,B9^2\,(B2 + B7 - B11)}{B5\,B6\,B11^2\,(B2 + B3 - B9)} \qquad \frac{B1\,B2\,+\,B1\,B3\,B6\,+\,B1\,B3\,B9\,+\,B3\,B4\,B5}{B11} \qquad (7)$$

Regarding the popularity of the different satellite bands in the evolved hyperfeatures, Figure 5 shows, for each band and each dataset, the fraction of hyperfeatures generated for that dataset (in 30 runs) that use the band. We only check whether a band appears in a hyperfeature. Measuring its importance inside the hyperfeature would be a complex exercise that we do not perform here. For each dataset, we subjectively identify a group of most popular bands as the ones ranked higher and at a larger distance from the rest. We do not identify any popular bands for Ao8, since on this dataset all the bands are ranked low, with small distances between them.

**In Binary Classification Datasets:** The most popular band in all four datasets was the SWIR2 (B7 in both LS-7 and LS-8), which appears in 62.4% to 81.4% of the hyperfeatures across all datasets. This preference for the SWIR2 band is expected due to its usefulness when searching for dry earth, which may indicate a recent fire [82].

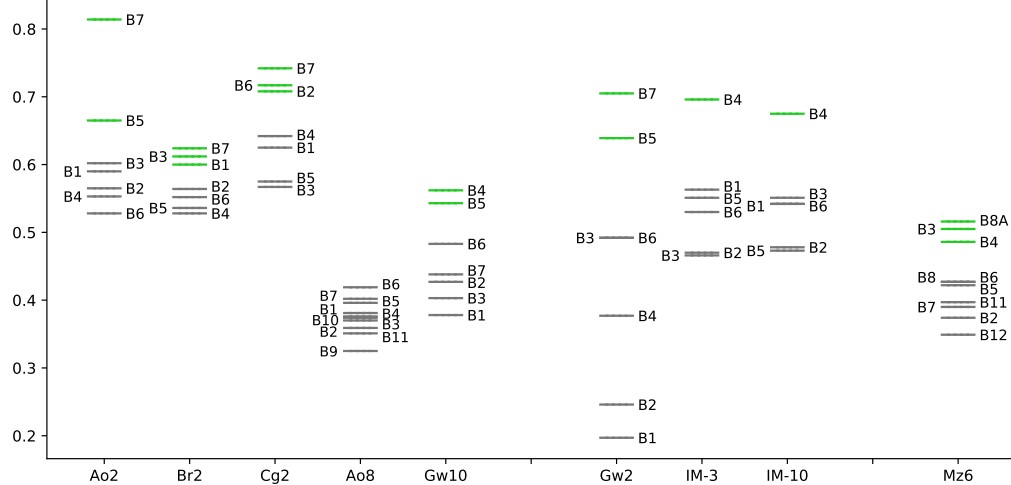

**Figure 5.** Fraction of hyperfeatures generated for each dataset (in 30 runs) that use a given band. The bands identified as popular are highlighted.

**When Discriminating Similar Forest Classes:** The most popular band in the IM-3 and IM-10 datasets was the NIR (B4), which appeared in 69.6% and 67.5% of the hyperfeatures, respectively. The popularity of the NIR band in the creation of hyperfeatures can be justified by its importance on the visualization of healthy vegetation in the discrimination between *Dense Forest* and *Open Forest* pixels. The importance of this band has also led to the creation of indices, such as the NDVI [1,82].

**When Discriminating All Classes:** Since the target classes are not similar to each other, here, we observe which are the most popular bands on each dataset, and attempt to explain their popularity based on which classes benefited the most from the hyperfeatures. We do not discuss the Ao8 dataset, not only because it lacks popular bands, but also because it did not benefit from the hyperfeatures, as we will see below in Section 4.4.

In the Gw10 dataset, the Red (B4) and NIR (B5) bands were the most popular, appearing in 56.2% and 54.3% of the hyperfeatures, respectively. We will see in the next section that on this dataset the hyperfeatures improved the classification of the *Mangrove, Savanna Woodland*, and *Wetland* pixels. This suggests that the better discrimination of these land cover types took into account the amount of healthy vegetation and the composition of the soil.

Regarding the Mz6 dataset, the Vegetation Red Edge (B8A), Green (B3), and Red (B4) were identified as the most popular bands, appearing in 51.6%, 50.5%, and 48.6% of the hyperfeatures, respectively. However, their effect is not so clear, since the improvement brought by the hyperfeatures affected several different classes. Taking into consideration only the classes with the highest improvement, which were *Agriculture/Bare Soil, Forest* and *Wetlands*, we suggest that the better discrimination of these land cover types considered the health and age of the vegetation, as well as the composition of the soil.

In some of the related work regarding the use of GP to build hyperfeatures in RS, the authors reveal what were considered the best hyperfeatures obtained. For comparison with our own, here we also comment on those hyperfeatures. It is important to say that these works address regression problems (rather than classification problems), which sometimes require more complex models in order to be solved. In comparison with our own, these works use an extensive list of mathematical operators to combine the original features (which also tends to cause the creation of larger models). The authors also include indices in the datasets, similarly to what is done in part of our work.

In [65], the authors use GP to monitor the quality of the water in reservoirs, by predicting the amount of chlorophyll in the water. The final model is quite simple (having a size of 8), according to our criteria, and only uses the Green, Red, and NIR bands. While in this case, the hyperfeature used is simple, that is not the case in the other two works. In [66], the authors attempt to predict the soil salinity. Their final model uses one band and five indices, and its size is near 50 (making it larger than any of our hyperfeatures). In [62], the authors attempt to predict the soil moisture and, although their final model only uses four terminals (SAR backscatter coefficient, slope, soil permeability (in/hr), and the NDVI), this model is the most complex out of these three.

### 4.2. Hyperfeatures in Binary Classification Datasets

The results obtained on the binary classification datasets (Ao2, Br2, Cd2, Gw2) are reported in Table 5 and Figure 6. In terms of training accuracy, the three classification algorithms managed to obtain perfect results in nearly every run, and therefore those results are not included in the table. In terms of test accuracy, the induced models achieved high values, nearly all above 99% (both in terms of overall accuracy and class accuracy, as seen in Table 6), also on the original (non-expanded) datasets. The lowest results belong to DT that, when applied to the Br2 dataset, achieved a median overall test accuracy of 98.9%. Without much room for improvement, FFX was still able to create hyperfeatures that improved the test accuracy in two cases (DT and XGB in the Gw2 dataset), surpassing also the M3GP hyperfeatures, while neither the indices nor the M3GP or EFS hyperfeatures caused any significant difference in the results. The boxplots show a low dispersion of

accuracy values (the ranges of the *y*-axes are limited), which seems to be marginally larger for the EFS results.

**Table 5.** Comparison of the median overall test accuracy (highlighted in bold) obtained by the three ML algorithms in the original datasets, when adding indices, and when using hyperfeatures evolved by EFS, FFX, and M3GP. The colored *p*-values indicate significantly <u>better</u>/*worse* results.

| Dataset | Decision Trees | | | | Random Forests | | | | XGBoost | | | |
|---|---|---|---|---|---|---|---|---|---|---|---|---|
| | Ao2 | Br2 | Cd2 | Gw2 | Ao2 | Br2 | Cd2 | Gw2 | Ao2 | Br2 | Cd2 | Gw2 |
| Orig. Dataset | | | | | | | | | | | | |
| **Test Accuracy** | **0.999** | **0.989** | **0.996** | **0.999** | **0.999** | **0.992** | **0.999** | **1.000** | **0.999** | **0.993** | **0.998** | **0.999** |
| Indices | | | | | | | | | | | | |
| **Test Accuracy** | **0.999** | **0.990** | **0.996** | **0.999** | **0.999** | **0.993** | **0.999** | **1.000** | **0.999** | **0.993** | **0.998** | **0.999** |
| *p*-value vs. Orig. | 0.694 | 0.678 | 0.909 | 0.669 | 0.675 | 0.917 | 0.436 | 0.871 | 1.000 | 1.000 | 1.000 | 1.000 |
| EFS | | | | | | | | | | | | |
| **Test Accuracy** | **0.998** | **0.989** | **0.996** | **0.999** | **1.000** | **0.992** | **0.999** | **1.000** | **0.999** | **0.993** | **0.998** | **0.999** |
| *p*-value vs. Orig. | 0.012 | 0.226 | 0.143 | 0.735 | 0.091 | 0.777 | 0.137 | 0.619 | 0.256 | 0.682 | 0.489 | 0.127 |
| FFX | | | | | | | | | | | | |
| **Test Accuracy** | **0.999** | **0.990** | **0.996** | **1.000** | **0.999** | **0.993** | **0.999** | **1.000** | **0.999** | **0.993** | **0.998** | **1.000** |
| *p*-value vs. Orig. | 0.224 | 0.941 | 0.294 | <u>0.000</u> | 0.363 | 0.988 | 0.148 | 0.730 | 0.739 | 0.794 | 0.886 | <u>0.000</u> |
| M3GP | | | | | | | | | | | | |
| **Test Accuracy** | **0.999** | **0.990** | **0.996** | **0.999** | **0.999** | **0.992** | **0.999** | **1.000** | **0.999** | **0.993** | **0.998** | **0.999** |
| *p*-value vs. Orig. | 0.908 | 0.947 | 0.672 | 0.813 | 0.500 | 0.846 | 0.688 | 0.871 | 1.000 | 1.000 | 1.000 | 1.000 |
| *p*-value vs. Ind. | 0.782 | 0.761 | 0.598 | 0.849 | 0.780 | 0.982 | 0.738 | 1.000 | 1.000 | 1.000 | 1.000 | 1.000 |
| *p*-value vs. FFX | 0.276 | 0.830 | 0.525 | *0.001* | 0.095 | 0.905 | 0.319 | 0.868 | 0.739 | 0.794 | 0.886 | *0.000* |
| *p*-value vs. EFS | 0.017 | 0.272 | 0.291 | 0.572 | 0.286 | 0.682 | 0.309 | 0.757 | 0.256 | 0.682 | 0.489 | 0.127 |

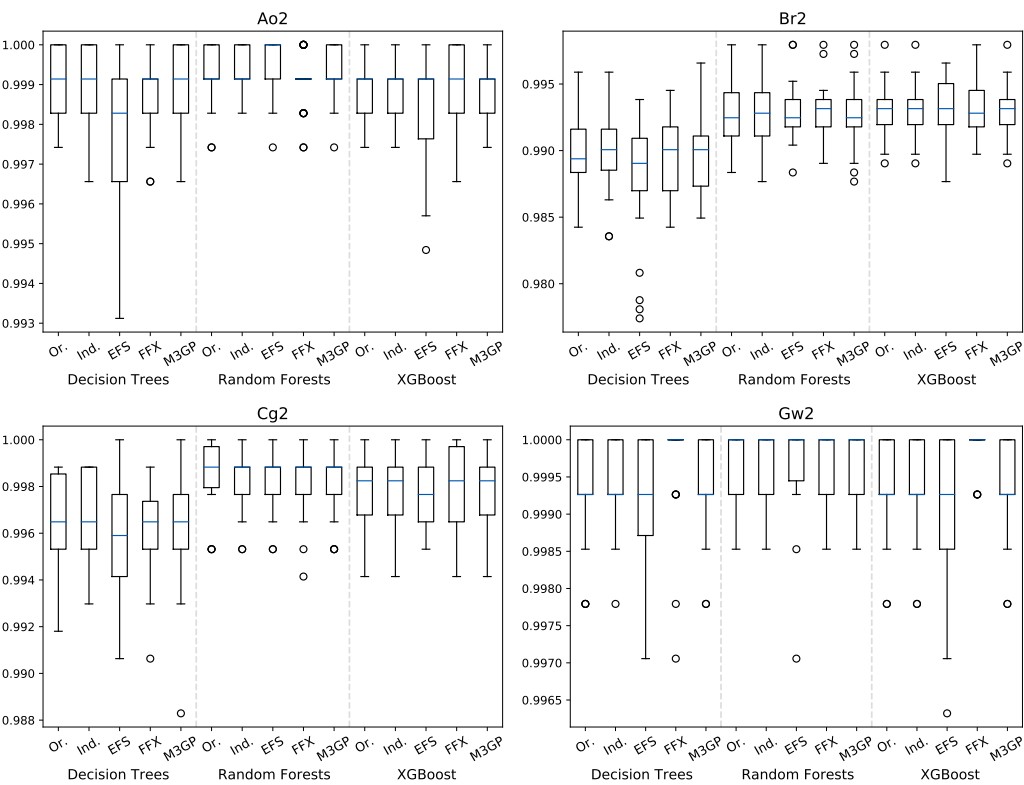

**Figure 6.** Boxplots of the test accuracy obtained in the binary classification datasets in each test case.

These seemingly uninteresting results agree with the findings of our previous work [11]. Using a different method of selecting and using the hyperfeatures also had no effect in the cases where the training and test datasets came from the same image. However, the hyperfeatures revealed to be beneficial when the induced models where applied to datasets that came from images not seen during training. This suggests that the current method of

obtaining and using the hyperfeatures may also prove beneficial in a similar training and test setting.

**Table 6.** Average test accuracy in each class when using the XGBoost algorithm in the original datasets.

| XGB—Original | Ao2 | Br2 | Cg2 | Gw2 | IM-3 | IM-10 | Gw10 | Ao8 | Mz6 |
|---|---|---|---|---|---|---|---|---|---|
| Agriculture/Bare Soil | — | — | — | — | — | 98.96% | 96.34% | 83.02% | 72.59% |
| Burnt | 99.81% | 99.51% | 99.63% | 99.80% | — | 93.19% | 98.72% | 99.22% | — |
| Clouds | — | — | — | — | — | — | — | 100.00% | — |
| Forest | — | — | — | — | — | — | — | 99.87% | 88.05% |
| - Dense Forest | — | — | — | — | 92.67% | 87.67% | 79.81% | — | — |
| - Open Forest | — | — | — | — | 93.02% | 93.65% | 98.41% | — | — |
| Grassland | — | — | — | — | — | 92.73% | 81.67% | 85.56% | 64.02% |
| Mangrove | — | — | — | — | — | 99.12% | 98.41% | — | — |
| Mud | — | — | — | — | — | 96.07% | — | — | — |
| Sand | — | — | — | — | — | 95.78% | 90.22% | — | — |
| Savanna Woodland | — | — | — | — | 99.71% | 84.41% | 98.66% | 98.66% | — |
| Urban | — | — | — | — | — | — | — | 87.78% | 56.82% |
| Water | — | — | — | — | — | 97.33% | 99.48% | 99.48% | — |
| Wetland | — | — | — | — | — | — | 95.29% | — | 80.34% |
| Other | 99.97% | 99.16% | 99.88% | 99.98% | — | — | — | — | 76.07% |

*4.3. Hyperfeatures to Discriminate Similar Classes in a Multiclass Classification Dataset*

Before looking at these results, it is worth recalling that the IM-3 dataset was built from three similar classes within the IM-10 dataset. As such, even though it has a reduced number of classes, it is not unexpected to see a lower accuracy in this dataset. It is also worth specifying that the hyperfeatures used in the IM-10 dataset were obtained only in the IM-3 dataset, in an attempt to help discriminate these similar classes. Finally, we also recall that EFS and FFX are not used in the multiclass datasets.

The results for IM-10 and IM-3 are reported in Table 7 and Figure 7. Once again, the training results are omitted from the table because all three algorithms achieved perfect results in nearly every run. In terms of test accuracy, we observe that, although the values are high, they have a larger margin for improvement when compared to the binary classification results reported above. When adding indices to the original dataset, the test accuracy on the IM-10 dataset increased with two algorithms (RF and XGB). When adding the hyperfeatures evolved by M3GP, the test accuracy in the IM-10 dataset increased with all three algorithms, and in the IM-3 dataset, it increased with the XGB algorithm. Neither the indices nor the M3GP hyperfeatures degraded the test accuracy. When comparing the performance of indices versus M3GP hyperfeatures, M3GP is better with two algorithms (DT and XGB). In the boxplots, we observe that IM-3 has a larger dispersion of values than IM-10 (notice the different $y$-axes ranges). On IM-10, the DT algorithm visibly falls behind RF and XGB.

**Table 7.** Comparison of the median overall test accuracy (highlighted in bold) obtained by the three ML algorithms in the original datasets, when adding indices, and when adding hyperfeatures evolved by the M3GP algorithm. The colored $p$-values indicate significantly better results.

| | Decision Trees | | Random Forests | | XGBoost | |
|---|---|---|---|---|---|---|
| **Dataset** | **IM-3** | **IM-10** | **IM-3** | **IM-10** | **IM-3** | **IM-10** |
| Original Dataset | | | | | | |
| **Test Accuracy** | **0.948** | **0.956** | **0.969** | **0.974** | **0.958** | **0.973** |
| Indices | | | | | | |
| **Test Accuracy** | **0.938** | **0.959** | **0.958** | **0.979** | **0.948** | **0.977** |
| $p$-value vs. Original | 0.151 | 0.051 | 0.062 | 0.000 | 0.178 | 0.001 |
| M3GP | | | | | | |
| **Test Accuracy** | **0.958** | **0.961** | **0.969** | **0.978** | **0.969** | **0.978** |
| $p$-value vs. Original | 0.020 | 0.000 | 0.844 | 0.000 | 0.009 | 0.000 |
| $p$-value vs. Indices | 0.000 | 0.407 | 0.055 | 0.218 | 0.000 | 0.711 |

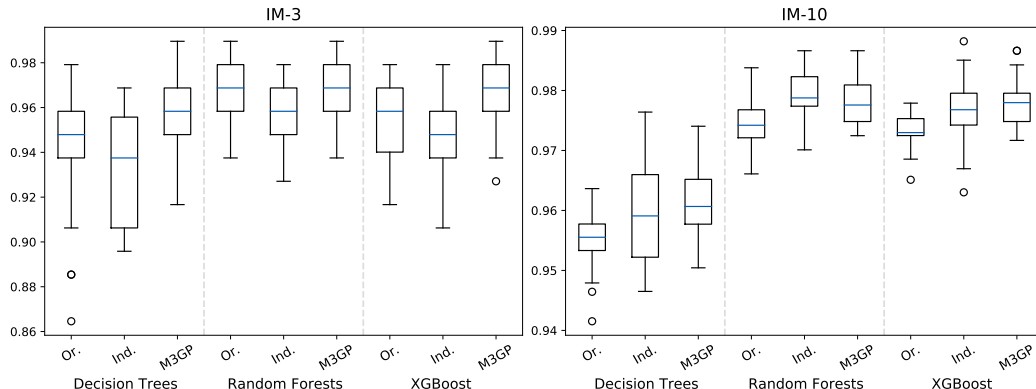

**Figure 7.** Boxplots of the test accuracy obtained in the IM-3 and IM-10 datasets in each test case.

In terms of class accuracy in the IM-3 dataset, we can see in Table 8 that the hyperfeatures improved the discrimination between the *Dense Forest* and the *Open Forest* pixels, at the cost of reducing the accuracy on the *Savanna Woodland* class, by increasing its confusion with *Open Forest*. Looking at Table 6, we see that *Savanna Woodland* had almost perfect accuracy, and therefore any changes on this class would certainly be for the worse. In the end, the three classes became more balanced in terms of accuracy. Although these hyperfeatures do not seem to be helpful in the classification of *Savanna Woodland* on the IM-3 dataset, when applied to the IM-10 dataset (See Table 9), their largest impact is precisely in this class by correcting pixels that were previously misclassified as *Agriculture/Bare soil*, *Grassland*, and *Mangrove*. Their second biggest impact is in the classification of *Dense Forest* by improving its discrimination from *Open Forest* and by correcting pixels that were previously misclassified as *Mangrove*.

**Table 8.** Confusion matrix comparing the average test accuracy obtained by the XGBoost algorithm with and without hyperfeatures in the IM-3 dataset. This table shows the difference in the percentage of pixels in each line. Improvements are in green and deteriorations in red. The rightmost column indicates the accuracy obtained in each class without using hyperfeatures and the coloured cells show the class accuracy difference from this value.

| XGB IM−3 | Dense Forest | Open Forest | Savanna Woodland | Original Accuracy |
|---|---|---|---|---|
| Dense Forest | 2.83% | −2.83% | 0.00% | 92.67% |
| Open Forest | −3.33% | 2.46% | 0.87% | 93.02% |
| Savanna Woodland | 0.00% | 1.57% | −1.57% | 99.71% |

**Table 9.** Confusion matrix comparing the average test accuracy obtained by the XGB with and without hyperfeatures in the IM-10 dataset. This table shows the difference in the percentage of pixels in each line. Improvements are in green and deteriorations in red. Only the 20 cells with the highest impact are colored. The rightmost column indicates the accuracy obtained without using hyperfeatures and the coloured cells show the class accuracy difference from this value.

| XGB IM-10 | Water | Burnt | Sand | Agriculture /Bare soil | Open Forest | Dense Forest | Grassland | Mangrove | Savanna Woodland | Mud | Original Accuracy |
|---|---|---|---|---|---|---|---|---|---|---|---|
| Water | 0.00% | −0.05% | 0.00% | 0.02% | 0.00% | 0.00% | 0.00% | −0.01% | 0.00% | 0.04% | 97.33% |
| Burnt | 0.87% | 0.29% | 0.00% | −0.29% | 0.00% | 0.00% | 0.14% | −0.87% | −0.58% | 0.43% | 93.19% |
| Sand | 0.00% | 0.00% | 0.95% | −0.95% | 0.00% | 0.00% | 0.00% | 0.00% | 0.00% | 0.00% | 95.78% |
| Agriculture /Bare soil | 0.00% | 0.00% | −0.15% | 0.19% | 0.00% | 0.00% | −0.01% | −0.18% | 0.13% | 0.02% | 98.96% |
| Open Forest | 0.00% | 0.00% | 0.00% | −0.08% | 1.03% | −1.83% | −0.08% | 0.79% | 0.16% | 0.00% | 93.65% |
| Dense Forest | 0.00% | 0.00% | 0.00% | 0.00% | −2.50% | 4.33% | 0.00% | −1.83% | 0.00% | 0.00% | 87.67% |
| Grassland | −0.15% | 0.15% | 0.00% | −0.15% | −0.45% | 0.00% | 0.15% | 0.61% | −0.15% | 0.00% | 92.73% |
| Mangrove | −0.15% | 0.00% | 0.00% | −0.03% | −0.03% | 0.00% | 0.00% | 0.23% | −0.04% | 0.01% | 99.12% |
| Savanna Woodland | 0.00% | −0.29% | 0.00% | −2.55% | 0.10% | 0.00% | −1.27% | −1.18% | 5.49% | −0.29% | 84.41% |
| Mud | −1.29% | 0.09% | 0.00% | −0.02% | 0.00% | 0.00% | 0.00% | 0.07% | 0.00% | 1.16% | 96.07% |

In these two datasets, although the M3GP hyperfeatures performed better than the indices, these were also clearly beneficial when added to the original datasets. This behavior was similar to all three classification algorithms and, as such, we only displayed the confusion matrices related to the XGBoost algorithm, which had the best results. It is worth noticing that the IM-3 and IM-10 datasets were extracted from a set of satellite images with different acquisition dates. Next, we will observe additional evidence that indices and hyperfeatures seem to be more useful in datasets coming from sets of images with different acquisition dates.

### 4.4. Hyperfeatures to Discriminate All Classes in Multiclass Classification Datasets

The results obtained on the three unrelated multiclass classification datasets (Ao8, Gw10, Mz6) are reported in Table 10 and Figure 8. Once again, the training results were omitted from the table, as perfect accuracy was achieved in almost every run. However, for the Mz6 dataset, XGB required a maximum tree depth larger than the implementation default in order to achieve it (see Section 3.5).

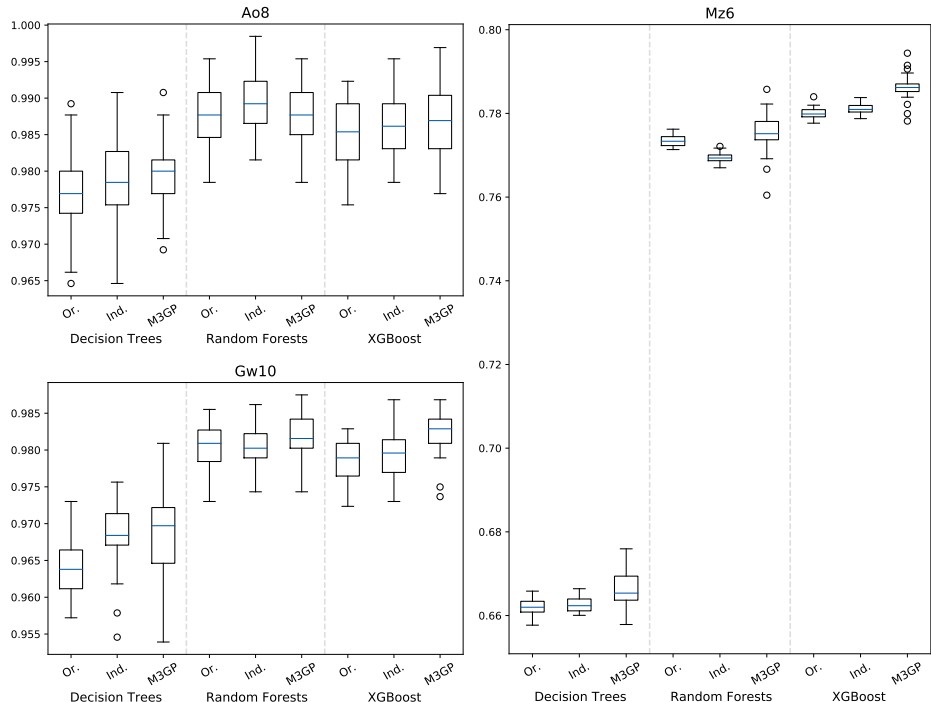

**Figure 8.** Boxplots of the test accuracy obtained in the Ao8, Gw10, and Mz6 datasets in each test case.

In terms of overall test accuracy, the indices improved the accuracy in two test cases (DT on Gw10, XGB on Mz6) and reduced the accuracy in one test case (RF on Mz6). On the other hand, the hyperfeatures evolved by M3GP improved the test accuracy in five test cases (Mz6 with all algorithms, and Gw10 with RF and XGB), when comparing the results with those on the original dataset, and in four test cases, when comparing with the results obtained with the indices (Mz6 with all algorithms, Gw10 with XGB). Once again, the hyperfeatures evolved by M3GP did not lead to a degradation of the test accuracy in any of the cases.

Both the indices and the M3GP-evolved hyperfeatures had an impact on the Gw10 and Mz6 datasets, which were obtained from a set of satellite images with different acquisition dates. Neither the indices nor the hyperfeatures had an impact on the Ao8 dataset, which was obtained from two images with the same acquisition date. These results, together with those displayed previously, seem to indicate that both the indices and the hyperfeatures are particularly useful in datasets obtained by mixing satellite images with different acquisition dates.

On the boxplots, once again we observe that DT falls behind RF and XGB, and completely struggles on the Mz6 problem.

**Table 10.** Comparison of the overall test accuracy (highlighted in bold) obtained by the three ML algorithms in the original datasets, when adding indices, and when adding hyperfeatures evolved by the M3GP algorithm. The colored *p*-values indicate significantly better/*worse* results.

| | Decision Trees | | | Random Forests | | | XGBoost | | |
|---|---|---|---|---|---|---|---|---|---|
| Dataset | Ao8 | Gw10 | Mz6 | Ao8 | Gw10 | Mz6 | Ao8 | Gw10 | Mz6 |
| Original Dataset | | | | | | | | | |
| **Test Accuracy** | **0.977** | **0.964** | **0.662** | **0.988** | **0.981** | **0.773** | **0.985** | **0.979** | **0.780** |
| Indices | | | | | | | | | |
| **Test Accuracy** | **0.978** | **0.968** | **0.662** | **0.989** | **0.980** | **0.769** | **0.986** | **0.980** | **0.781** |
| *p*-value vs. Original | 0.291 | 0.000 | 0.371 | 0.213 | 0.824 | *0.000* | 0.645 | 0.335 | 0.003 |
| M3GP | | | | | | | | | |
| **Test Accuracy** | **0.980** | **0.970** | **0.665** | **0.988** | **0.982** | **0.775** | **0.987** | **0.983** | **0.786** |
| *p*-value vs. Original | 0.125 | 0.000 | 0.000 | 0.923 | 0.054 | 0.006 | 0.389 | 0.000 | 0.000 |
| *p*-value vs. Indices | 0.693 | 0.847 | 0.000 | 0.228 | 0.038 | 0.000 | 0.650 | 0.002 | 0.000 |

In terms of class accuracy in the Gw10 dataset (Table 11), when using the XGBoost algorithm, the improvements are more general across the classes, with the exception of the *Grassland* pixels, which are now misclassified as *Savanna Woodland*, and the *Dense Forest* pixels, which were previously misclassified as *Open Forest*.

In this case, we omitted the results regarding the RF classifier since there was no statistically significant difference between the runs in the original and the extended datasets. We present the results of the XGBoost classifier to show that it is also possible to create a set of hyperfeatures that helps improving the accuracy on nearly all classes.

Regarding class accuracy in the Mz6 case (Table 12), when using the hyperfeatures in the XGBoost algorithm, the improvement was general among all classes, with a higher impact on the *Urban* pixels that were previously misclassified as *Other*. The improvements in this class's pixels can be easily justified by being the class with the lowest accuracy in the original dataset, followed by *Grassland*, which was also improved.

**Table 11.** Confusion matrix comparing the average test accuracy obtained by the XGB with and without hyperfeatures in the Gw10 dataset. This table shows the difference in the percentage of pixels in each line. Improvements are in green and deteriorations in *red*. Only the 20 cells with the highest impact are colored. The rightmost column indicates the accuracy obtained without using hyperfeatures and the coloured cells show the class accuracy difference from this value.

| XGB Gw10 | Agriculture /Bare soil | Burnt | Dense Forest | Grassland | Mangrove | Open Forest | Sand | Savanna Woodland | Water | Wetland | Original Accuracy |
|---|---|---|---|---|---|---|---|---|---|---|---|
| Agriculture /Bare soil | 0.77% | 0.00% | 0.00% | 0.00% | 0.00% | 0.00% | −0.62% | −0.17% | 0.00% | 0.02% | 96.34% |
| Burnt | −0.14% | 0.28% | 0.00% | 0.00% | −0.43% | −0.07% | 0.00% | 0.21% | 0.00% | 0.14% | 98.72% |
| Dense Forest | 0.00% | 0.00% | 0.93% | 0.00% | 0.93% | −1.85% | 0.00% | 0.00% | 0.00% | 0.00% | 79.81% |
| Grassland | 0.00% | 0.00% | 0.00% | −2.50% | 0.00% | 0.00% | 0.00% | 2.50% | 0.00% | 0.00% | 81.67% |
| Mangrove | 0.00% | 0.00% | −0.03% | 0.00% | 0.40% | −0.02% | 0.00% | −0.08% | −0.08% | −0.19% | 98.41% |
| Open Forest | 0.00% | 0.00% | −0.19% | 0.00% | −0.03% | 0.31% | 0.00% | −0.09% | 0.00% | 0.00% | 98.41% |
| Sand | 0.67% | 0.00% | 0.00% | 0.00% | 0.00% | 0.00% | −0.67% | 0.00% | 0.00% | 0.00% | 90.22% |
| Savanna Woodland | −0.09% | 0.00% | 0.00% | 0.00% | −0.02% | −0.14% | 0.00% | 0.29% | 0.00% | −0.04% | 98.66% |
| Water | 0.00% | 0.00% | 0.00% | 0.00% | −0.05% | 0.00% | 0.00% | 0.00% | 0.16% | −0.11% | 99.48% |
| Wetland | −0.11% | 0.06% | 0.00% | 0.00% | −0.34% | 0.00% | 0.00% | 0.00% | 0.06% | 0.34% | 95.29% |

**Table 12.** Confusion matrix comparing the average test accuracy obtained by the XGBoost algorithm with and without hyperfeatures in the Mz6 dataset. This table shows the difference in the percentage of pixels in each line. Improvements are shown in green. The 10 cells with the highest impact are colored. The rightmost column indicates the accuracy obtained in each class without using hyperfeatures and the coloured cells show the class accuracy difference from this value.

| XGB-Mz6 | Agriculture /Bare soil | Forest | Grassland | Urban | Wetland | Other | Original Accuracy |
|---|---|---|---|---|---|---|---|
| Agriculture /Bare soil | 0.87% | −0.24% | −0.28% | −0.01% | −0.01% | −0.32% | 72.59% |
| Forest | −0.00% | 0.34% | −0.29% | 0.00% | −0.04% | −0.01% | 88.05% |
| Grassland | −0.26% | −0.32% | 0.51% | 0.00% | 0.14% | −0.08% | 64.02% |
| Urban | −0.03% | −0.02% | −0.25% | 1.82% | −0.06% | −1.48% | 56.82% |
| Wetland | −0.15% | −0.19% | −0.39% | 0.02% | 0.84% | −0.12% | 80.34% |
| Other | −0.26% | −0.04% | −0.18% | −0.04% | −0.08% | 0.60% | 76.07% |

*4.5. Impact on the MRV Performance*

When training hyperfeatures to discriminate multiple classes, the results indicate an overall improvement, particularly in the classes that previously had a lower accuracy. The improvements are significant in the IM-3, IM-10, Gw10, and Mz6 datasets. These datasets have one thing in common: they consist of mosaics derived from several acquisition dates. On the contrary, in the Ao8 dataset, where the two images of the mosaic are from the same day, there are no significant improvements. These results indicate that both indices and hyperfeatures are more useful when training models in images with more than one acquisition date (or from different locations). However, when this is not the case, there is no visible degradation of accuracy either.

Monitoring forest land cover at the country level, in compliance with UNFCCC standards, is a challenging endeavor, especially for vast countries covering various ecosystem types with distinct seasonality. The production of wall-to-wall maps derived from satellite imagery is especially attractive in these cases because remote sensing can cover large extents, greatly reducing costs, improving consistency, and increasing the periodicity of observations. However, in such cases, the image mosaics required to produce good quality maps are likely to include many different acquisition dates, maximizing image quality and observation date adequacy regarding vegetation cycles and climatic conditions. Thus, considering the results obtained, with hyperfeatures improving both the discrimination of analogous wooded vegetation classes and the classification accuracy of large image mosaics, with no degradation of results in any of the cases tested, it can be ascertained that the methods presented merit further development to exploit improvements in remote-sensing-based MRV performance. They are also general enough to be applied to a variety of other tasks, inside and outside the remote sensing domain.

**5. Conclusions**

We performed feature construction using M3GP, a variant of the standard genetic programming algorithm, with the goal of improving the performance of several machine learning algorithms by adding the new hyperfeatures to the reference datasets. We tested the approach in the tasks of binary classification of burnt areas and multiclass classification of land cover types. The datasets used were obtained from Landsat-7, Landsat-8, and Sentinel-2A satellite images over the countries of Angola, Brazil, Democratic Republic of Congo, Guinea-Bissau, and Mozambique.

The hyperfeatures produced by the M3GP algorithm, although variable in number and size, were generally not complex and were considered to be quite interpretable. While a larger number of hyperfeatures were created on the multiclass classification problems, a higher dispersion of sizes was observed on the binary problems. Regarding the popularity of each satellite band in the binary and multiclass classification problems, the models frequently used the SWIR2 band when trying to detect burnt areas in the binary datasets. On the multiclass classification datasets, the models seemed to have a preference for the Vegetation Red Edge,

NIR, Red, and Green bands when training hyperfeatures to discriminate different forest classes or when the hardest classes included vegetation (e.g., *Agriculture/Bare Soil* and *Forest*), and in some cases, also water (e.g., *Mangroves* and *Wetlands*).

The performance of decision trees, random forests, and XGBoost was assessed on the original datasets and on the datasets expanded with the evolved hyperfeatures, and the results compared for statistical significance. For comparison purposes, we also assessed the performance of the same algorithms on all datasets expanded with the well-known spectral indices NDVI, NDWI, and NBR, and on the binary datasets expanded with hyperfeatures created by the FFX and EFS feature construction algorithms. On the binary classification problems, we conclude that neither of the four alternatives (M3GP, indices, FFX, EFS) leads to substantial improvements. Only FFX was able to improve the results in 2 out of 12 test cases (both on the same dataset). On the multiclass classification problems, the hyperfeatures evolved by the M3GP caused significant improvements in 9 out of 15 test cases, with no degradation of results in any test case, while the indices caused significant improvements in 4 out of 15 test cases and significant degradation of results in one test case. The approach appears to be equally beneficial to all three machine learning algorithms.

Overall, both hyperfeatures and indices displayed the capability of improving the robustness of the machine learning models in multiclass classification datasets. However, this improvement seems to exist only in datasets built from collections of images with several acquisition dates, which indicates that both hyperfeatures and indices can be robust to the radiometric variations across images and can be used to improve the MRV performance of mechanisms such as REDD+.

Although the hyperfeatures have the advantage of being created automatically with specific goals, such as the discrimination of specific classes, there is a computational cost associated with this task. Taking this into consideration, one of our objectives for future work is to continue the validation of the efficacy of the hyperfeatures in the discrimination of similar classes and their robustness to the radiometric variations across different satellite images. We hope to be able to create reusable hyperfeatures, thus reducing the computational cost of generating them frequently. Besides this validation, we also want to expand this work into regression problems, such as the estimation of biomass from satellite images.

**Author Contributions:** Conceptualization, J.E.B. and S.S.; methodology, J.E.B.; software, J.E.B.; validation, S.S.; formal analysis, J.E.B.; investigation, J.E.B.; resources, S.S.; data curation, A.I.R.C.; writing—original draft preparation, J.B and L.V.; writing—review and editing, A.I.R.C., J.E.B., M.J.P.V., and S.S.; visualization, A.I.R.C. and J.E.B.; supervision, S.S.; project administration, S.S.; funding acquisition, L.V., M.J.P.V., and S.S. All authors have read and agreed to the published version of the manuscript.

**Funding:** This work was partially supported by FCT through funding of LASIGE Research Unit (UIDB/00408/2020 and UIDP/00408/2020) and CEF (UIDB/00239/2020); projects BINDER (PTDC/CCI-INF/29168/2017), OPTOX (PTDC/CTA-AMB/30056/2017), PREDICT (PTDC/CCI-CIF/29877/2017), INTERPHENO (PTDC/ASP-PLA/28726/2017), GADgET (DSAIPA/DS/0022/2018), AICE (DSAIPA/DS/0113/2019); PhD Grant (SFRH/BD/143972/2019).

**Data Availability Statement:** The datasets used in this work are available at https://github.com/jespb/RSJ_21, except for the Mz6 dataset which is proprietary.

**Conflicts of Interest:** The authors declare no conflict of interest. The funders had no role in the design of the study; in the collection, analyses, or interpretation of data; in the writing of the manuscript, or in the decision to publish the results.

## Abbreviations

The following abbreviations are used in this manuscript:

| | |
|---|---|
| Af | Equatorial rainforest, fully humid |
| Am | Equatorial monsoon |
| Ao | Angola |
| Aw | Equatorial savanna with dry winter |
| B$x$ | Band $x$ |
| Br | Brazil |
| BSh | Hot semiarid |
| Bwh | Hot desert |
| CCDC | Continuous change detection and classification |
| Cd | Democratic Republic of the Congo |
| Cwa | Warm temperate climate with dry winter and hot summer |
| Cwb | Warm temperate climate with dry winter and warm summer |
| DT | Decision tree |
| EC | Evolutionary computation |
| EFS | Evolutionary feature synthesis (algorithm) |
| FFX | Fast function extraction (algorithm) |
| GLCM | Gray level co-occurrence matrix |
| Gw | Guinea-Bissau |
| GP | Genetic programming |
| KGCS | Köpper–Geiger classification system |
| LS-7 | Landsat 7 |
| LS-8 | Landsat 8 |
| M3GP | Multidimensional multiclass GP with multidimensional populations (algorithm or classifier) |
| MD | Mahalanobis distance (classifier) |
| ML | Machine learning |
| MRV | Measure, report, and verify |
| Mz | Mozambique |
| NBR | Normalized burn ratio |
| NDVI | Normalized difference vegetation index |
| NDWI | Normalized difference water index |
| PCA | Principal component analysis |
| REDD+ | Reducing emissions from deforestation and forest degradation |
| RF | Random forest |
| RS | Remote sensing |
| S-2A | Sentinel-2A |
| UNFCCC | United Nations Framework Convention on Climate Change |
| XGB | XGBoost |
| WAF | Weighted average of $F$-measures |

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
