# Peer review of "Improving Land Cover Classification Using Genetic Programming for Feature Construction"

_remotesensing, doi:10.3390/rs13091623_

Round 1
Reviewer 1 Report
Dear Authors;
Your manuscript is too long and make the reader loss the main idea, your contribution is not clear, so many tables and too many data sets. your manuscript needs more trim to make it easy to be understandable.
1- Page 3 of 27, Line 108, you have to change the word and colleagues to et al.
2- Page 10 of 27, Line 335, one individual achieves 100% accuracy on the training set as a stop condition is very high, have you arrived this accuracy during your experiments?
Author Response
> Your manuscript is too long and make the reader loss the main idea, your contribution is not clear, so many tables and too many data sets. your manuscript needs more trim to make it easy to be understandable.
Response:
Thank you for your feedback. We agree that this document is long but, this length is necessary given the number of datasets and study areas in question. This number of datasets strengthens the conclusions by allowing us to make conclusions such as that the hyper-features were more useful in a particular kind of dataset (datasets obtained from sets of images with multiple acquisition dates).
Given that the other reviewers did not seem to be bothered by the length of the manuscript, we decided to carefully select what could be trimmed while maintaining the same main conclusions. In the original submission, the Conclusions section ended at the final of page 22. This was pushed back to the middle of page 21 after performing the modifications stated below:
Actions:
3. Material and Methods:
The introduction for this section did not give any non-obvious information about this section. Also taking into consideration the commentary by Reviewer #4, we removed this introduction.
4.1. M3GP Performance and Hyper-feature Analysis:
Table 5 and Figure 5 contained the same information in a different format. To promote a more light reading, we removed Table 5. The commentaries referring to the percentage values in Table 5 were maintained since these values are represented in Figure 5.
4.4. Hyper-features to Discriminate All Classes in Multiclass Classification Datasets:
The changes in the accuracy of the Decision Tree classifier in each class of the Gw10 dataset when using hyper-features, compared with not using hyper-features, are displayed in Table 12. Table 13 displays the equivalent information when using the XGBoost classifier. In terms of main conclusions, both tables indicate that the hyper-features improve the accuracy in most classes. However, the effects in each class are different in each classifier. Although these results are interesting and deserve to be explored, this work is not focused on the effects of the hyper-features in different classifiers. As such, we decided to remove Table 12 and keep Table 13, for consistency of showing the table referring to the state-of-the-art classifier XGBoost.
4.4. Hyper-features to Discriminate All Classes in Multiclass Classification Datasets:
The changes in the accuracy of the Random Forest classifier in each class of the Mz6 dataset when using indices, compared with not using indices, are displayed in Table 14. Table 15 displays the equivalent information when using hyper-features on the XGBoost classifier. The results displayed in Table 14 show that the accuracy decreased when using indices on the Random Forest classifier. Although these results are also interesting and deserve to be explored, the commentaries were outside the scope of the paper's main conclusions. Since the p-value tables also show this reduction in overall accuracy, we decided to remove Table 14 and focus the commentaries regarding this dataset on the results displayed in Table 15.
Notes about the tables removed from 4.4:
The main conclusion of this work is that hyper-features created by the M3GP seem to be useful in datasets obtained from sets of satellite images with multiple acquisition dates. In this aspect, the information within the Decision Tree table is redundant, since equivalent information is seen in the XGBoost table. Also in this aspect, the information within the Random Forest table is off-topic, since it displays information about accuracy reductions when using indices.
> Page 3 of 27, Line 108, you have to change the word and colleagues to et al.
Action: We corrected this, and another similar issue in the same paragraph, to "et al."
> Page 10 of 27, Line 335, one individual achieves 100% accuracy on the training set as a stop condition is very high, have you arrived this accuracy during your experiments?
Response: Since the fitness function has a maximum value of 100% accuracy, it makes sense to include this value in the stopping criteria. An early stop, in this case, avoids both the growth of the expression and spending unnecessary time in training models. Although this value is not usually reached in more complex datasets, it was reached sometimes in the later generations on the Ao8 and IM-3 datasets, in mid generations in the Ao2 dataset, and on early generations in the Gw2 dataset. In Table 4, you can see that the M3GP is able to obtain a median overall training accuracy greater than 99.5% in these 4 datasets.
Reviewer 2 Report
The article can be accepted after minor revision (corrections to minor methodological errors and text editing)
Author Response
Thank you for the positive review. We tried to correct minor grammatical issues while answering the reviewers concerns.
Reviewer 3 Report
The paper presented by the authors is of broad interest for the remote sensing community, the objective of the research and the results obtained proves of the overall performance of the methodology implemented. Moreover, the structure of the document, its redaction and explanations are clear. Therefore, I recommend the publication of this paper.
Author Response
Thank you for the positive review.
Reviewer 4 Report
The manuscript shows that adding the M3GP hyper-features to the reference datasets brings better results than adding the well-known spectral indices NDVI, NDWI and NBR by comparing the performance of the M3GP hyper-features in the binary classification problems with those created by other Feature Construction methods like FFX and EFS.
This is a very well-written manuscript. The introduction section includes all significant related information, sufficiently highlighting the importance of the topic and the novelty of this work. All parts of the methodology are described in a comprehensive manner. The results are valuable and are properly discussed.
Nonetheless, still some reviews should be performed.
Line 108: "....investigated by Zhang and colleagues". Citation?
Lines 109, 117, etc.: It's better to use Neshatian et al. [44] instead of "in [44]"
Line 128: It's better to rephrase "Different approaches based on GP to constructing multiple features were investigated [48]"
Lines 171-174: Is this really necessary?
Line 197: Why is dimension "m2" italicized?
Line 203: Correct 1500mm -> 1500 mm
Author Response
> The manuscript shows that adding the M3GP hyper-features to the reference datasets brings better results than adding the well-known spectral indices NDVI, NDWI and NBR by comparing the performance of the M3GP hyper-features in the binary classification problems with those created by other Feature Construction methods like FFX and EFS.
> This is a very well-written manuscript. The introduction section includes all significant related information, sufficiently highlighting the importance of the topic and the novelty of this work. All parts of the methodology are described in a comprehensive manner. The results are valuable and are properly discussed.
Thank you for your feedback. Every concern was taken into consideration when editing the newest version of this document.
> Nonetheless, still some reviews should be performed.
> Line 108: "....investigated by Zhang and colleagues". Citation?
This whole paragraph reported work investigated by Zhang et al., for that reason, we did not include the citation right at the beginning. However, we do agree that introducing the citations here improves the clarity of the paragraph.
The citations were added.
> Lines 109, 117, etc.: It's better to use Neshatian et al. [44] instead of "in [44]"
We agree that it is better to include the name of the authors in some cases (e.g., we edited line 79) and that this is one of those cases. However, since this paragraph is introduced as the work by Zhang et al., we believe that omitting the names produces a smoother flow of information.
We searched for other instances where "in [X]" was used and, if we considered that the citation should be highlighted, changed the phrase to include "Name et al. [X]". We only modified line 79.
> Line 128: It's better to rephrase "Different approaches based on GP to constructing multiple features were investigated [48]"
We agree that this phrase can be improved. We replaced "features" with "hyper-features" for consistency with the other sections of this work.
This phrase was modified to "In [48], the authors investigate the construction of sets of hyper-features using GP-based approaches."
> Lines 171-174: Is this really necessary?
This paragraph does not bring any non-obvious information about the section. As such, we decided to remove it.
> Line 197: Why is dimension "m2" italicized?
By mistake, the "m" was included in the latex expression used "m$^2$".
> Line 203: Correct 1500mm -> 1500 mm
This, and other instances where units of measurement are used, were corrected.
Round 2
Reviewer 1 Report
Dear Authors;
I'd like to thank you for this comprehensive modifications, which introduced a good article.